# Perinatal suicidal behavior in sub-Saharan Africa: A study protocol for a systematic review with meta-analysis

**Mohammedamin Hajure**[1]*, **Gebiso Roba**[2], **Wubishet Gezimu**[3], **Desalegn Nigatu**[3],
**Mustefa Mohammedhussein**[4], **Jemal Ebrahim**[4], **Aman Mamo**[1], **Aman Dule**[1], **Kiyar Jemal**[5]

**1** Department of Nursing, College of Health Sciences, Madda Walabu University, Shashemene, Ethiopia,
**2** Department of Public health, College of Health Sciences, Mattu University, Mattu, Ethiopia, **3** Department of Nursing, College of Health Sciences, Mattu University, Mattu, Ethiopia, **4** Department of psychiatry, College of Health Sciences, Madda Walabu University, Bale, Ethiopia, **5** Department of Medical Laboratory Science, College of Medicine and Health Sciences, Arba Minch University, Arba Minch, Ethiopia

* sikoado340@gmail.com

**Data Availability Statement:** The identified research data will be made publicly available when the study is completed and published.

## Abstract

### Background

Perinatal mental illnesses are predominant during gestation and continue for a year after delivery. According to the International Statistical Classification of Diseases and Related Health Problems, Tenth Revision (ICD-10), suicide is classified as a direct cause of death among the maternal population. The occurrence of suicidal behavior among perinatal women was considered the main contributor to the burden of the disorder. Hence, the current study will develop a protocol for a systematic review as well as a meta-analysis on estimating the prevalence and determinants of perinatal suicidal behavior in Sub-Saharan African countries.

### Methods

PubMed/MEDLINE, Scopus, EMBASE, PsycINFO, and the Web of Science electronic databases will be searched for studies reporting primary data. The second search strategy will be done with Google Scholar, using a combination of the medical subject headings and keywords as the search terms. The studies will be classified into included, excluded, and undecided categories. The studies will be judged based on the eligibility criteria. Heterogeneity will be checked by using the $I^2$ test (Cochran Q test) at a p-value of 0.05 and assuming that the $I^2$ value is > 50%. Publication bias will be checked using a funnel plot, Beg's rank, and Eggers linear statistical tests. A subgroup analysis and sensitivity test will be carried out. The risk of bias will be assessed using the Joanna Briggs Institute (JBI), and the quantitative analysis will determine whether or not to proceed based on the results.

### Discussion

This protocol's comprehensive review is expected to generate sufficient evidence on the prevalence of suicidal behavior and its determinants among women during the perinatal

**Funding:** The author(s) received no specific funding for this work.

**Competing interests:** The authors have declared that no competing interests exist.

period in Sub-Saharan African countries over the last two decades. Hence, this protocol will be imperative to collect and combine empirical data on suicidal behavior during the perinatal period, and doing so will help to provide essential implications or better evidence to plan different kinds of interventions considering determinants expected to impact the burden of suicidal behavior during the perinatal period.

## Systematic review registration

PROSPERO (CRD42022331544).

## Introduction

Perinatal mental illnesses are prevalent throughout pregnancy and last for up to a year following delivery. Suicide is one of several perinatal psychiatric symptoms that have been reported in the literature [1]. Suicidal behavior in women is thought to be one of the largest contributors to the burden of the condition in this subgroup of populations [2]. The perinatal period is a crucial time in a woman's life since it is linked to increased moodiness due to hormonal changes, which further increases the risk of suicidal behavior [3]. As per research, suicide is the fourth-leading cause of death worldwide among reproductive-aged women (15–49 years) [4] and the leading cause of death for young women in low-resource settings [3, 5, 6]. Around the world, suicide makes up roughly 1.7% of pregnancy-related deaths, with Southeast Asia having the highest prevalence (2.2%) [7]. According to the International Statistical Classification of Diseases and Related Health Problems, Tenth Revision (ICD-10), suicide is classified as a direct cause of death among the maternal population [8]. It accounted for almost 11% of deaths [9] and was found to be the second-leading cause of death in this subgroup altogether, with statistics showing that 20% of suicidal behavior occurs in other settings [10]. This was accompanied by the use of hazardous methods among this population, indicating a peak of suicidal intent and possibly defining a severe underlying psychiatric condition [11]. Studies have repeatedly demonstrated that mother-child suicide is the primary cause of death in both poor and high-resource settings [7, 11, 12] and is a commonly occurring feature that increases the risk of infanticide [12]. A WHO report shows more than 77% of suicides occur in low and middle-income countries like Sub-Saharan Africa [13] even if the concern of maternal suicide, typically during pregnancy, has been disregarded as it is not reported as a cause of death or defined as such [14].

The magnitude of the problem differs for types of suicidal behavior such as thoughts, plans, and attempts. Suicidal ideation was reported in perinatal women in various countries, ranging from 10.3% in Brazil [15], 11% in Pakistan [16], and 22.6% in Peru [17]. Furthermore, a report from South Africa [18], Egypt [19], and Ethiopia [20, 21] showed that about 18%, 20.4%, and 13.3%-53.2% of the respondents had current suicidal ideations, respectively. Similarly, a study found that a lifetime suicide attempt during the current pregnancy ranges from 1.8% in Egypt [19] to 13.3% in Brazil [22] and a suicidal plan was found to be 7.2% in Peru [17]. In general, suicidal ideation during pregnancy is estimated to be 12–21% in Africa [2, 18, 19], with literature indicating that the proportion of suicidal attempts during the perinatal period ranges from 1.8% in Egypt [19] to 78.3% in Ethiopia [23].

The magnitude of suicidal behavior in low- and middle-income countries varies among pregnant women. Accordingly, the magnitude of lifetime suicidal thoughts ranges from 11% to 78.3%, and the proportion of lifetime suicidal attempts varies from 2.7% to 36.2% in Ethiopia [20, 23]. The study also shows that postpartum women are more likely than antepartum

women to engage in suicidal behavior, with rates ranging from 4 to 17.6% and being more common in low-resource settings [24–27]. In contrast to this, studies reported a slightly higher preponderance of suicidal behavior during the antenatal period, ranging from 14% in Bahirdar to 47% in Gedeo, Ethiopia [23, 28].

During the perinatal period, women may have the chance to have regular follow-ups with health professionals, particularly in developed regions of the world, for prompt intervention, including in cases of suicidal behavior when they first report warning signs. However, as evidenced by Daniela C et al., whose systematic review and meta-analysis show a pooled prevalence rate of maternal deaths attributed to suicide ranging from 1.0% to 1.7% [7], prevention of suicidal behavior during the antenatal and postnatal period has been a missed opportunity in low-resource settings. Various factors, including demographic, psychosocial, cognitive, and clinical correlates, have been investigated in the literature as being linked with perinatal women's suicide behavior. These factors include depression [11, 28–31], anxiety and impulse control disorder [32, 33], being younger [34–36], unemployment and lower educational status, intimate partner violence [28, 37–39], cultural influence [40], social stigma [39], sleep disorders, khat chewing, alcohol, and tobacco use, economic crisis, poor social network [20, 35, 39, 41, 42], history of childhood abuse, rape, and verbal abuse [42–44], unwanted pregnancy [28, 41, 45], gestational age (GA) higher than 27 weeks [28], having an unfaithful husband [42], family stress [39, 46], poverty, chronic medical illness [23, 28, 36, 39], and previous trauma [45]. Moreover, attending treatment during the postpartum period was shown to reduce the suicide rate compared with age-matched women in the community. Additionally, it stated that, among pregnant women, suicidal thinking is the most typical predictor of subsequent suicidal attempts and completion [11, 38]. Significant evidence of this was provided in the years after the initial attempt to end oneself [35].

Suicidal behavior assessment must consider all domains of conditions that are directly or indirectly associated with an increased risk rather than being limited to terminological explanations such as thought, gesture, attempt, and commit. Numerous prior studies on suicide during the prenatal and postpartum periods have been conducted in developed countries, with sufficient output for potential implications in the use of strategies or policies. On the other hand, the effects of suicide behaviors among pregnant women in low-resource settings, such as Sub-Saharan countries, have received little attention in the empirical literature.

## Study objectives

### General objective.

- The aim of this systematic review and meta-analysis is to estimate the pooled prevalence of perinatal suicidal behavior, and identify its determinants in Sub-Saharan African countries.

### Specific objectives.

- To assess the prevalence of perinatal suicidal behavior in sub-Saharan African countries
- To identify the determinants of perinatal suicidal behavior in sub-Saharan African countries

## Research questions

**General question.** This protocol aims at answering the questions: what is the prevalence of perinatal suicide behavior in sub-Saharan African countries, as well as what determines that prevalence?

**Specific questions.**

- What is the prevalence of perinatal suicidal behavior in sub-Saharan African countries?

- What are the determinants of perinatal suicidal behavior in sub-Saharan African countries?

## Materials and methods

### Study design

The review will be done as per the guidelines stated in the Preferred Reporting Items for Systematic Reviews and Meta-analyses (PRISMA-2020) [47] (S1 Checklist) and registered on PROSPERO with the registration number (CRD42022331544).

### Risk of bias (quality) assessment

Two authors will assess the methodological quality of included studies using either the Joanna Briggs Institute Meta-analysis of Statistics Assessment and Review Instrument (JBI-MAStARI) [48] or the critical appraisal tool for prevalence studies and report as per the Preferred Reporting Items for Systematic Reviews and Meta-analyses (PRISMA). Any uncertainties will be resolved through mutual discussion and agreement.

### Definition of concepts

Perinatal period: The perinatal period is defined as the interval between conception and one year following childbirth.

Suicide: self-inflicted death with explicit or implicit evidence that the person intended to die.

Suicide intent: subjective expectation and desire for a self-destructive act to end in death.

Suicide attempt: self-injurious behavior with nonfatal outcome accompanied by explicit or implicit evidence that the person intended to die.

Suicide complete: serving as the agent of one's own death.

Suicidal behavior- includes ideation, suicidal intent, thought attempt and complete.

Lethality of suicidal behavior—objective danger to life associated with a suicide method or action [49].

### Eligibility criteria

The review will follow the Population, Exposure, Comparison, Outcome, and Study Design (PECOS) guideline. Accordingly, all perinatal (antenatal, intrapartum, and postpartum) women who lived in sub-Saharan Africa, regardless of their socio-demographic differences, obstetric characteristics, and healthcare service coverage and utilization, will be considered study participants (**P**). Exposure (**E**) to socio-demographic, socio-economic, obstetric, psychosocial, and other factors that influence suicidal behavior among perinatal women will be reviewed. Since the proposed review will assess the prevalence of perinatal suicidal behavior it will lack a comparison (**C**). The main primary outcome of the review, outcome (**O**), will be the pooled prevalence of perinatal suicidal behavior. Moreover, the review will include only peer-reviewed articles (**S**) published in English. All quantitative observational studies (cross-sectional, case-control, cohort, or longitudinal studies, and survey findings) and randomized control trial (RCT) studies will be included. The review will include all full-length articles reporting the prevalence, magnitude, and factors associated with perinatal suicidal behavior. However, all types of qualitative studies, case reports, commentaries, reviews, editorials, and

conference abstracts with inadequate information will be excluded. To avoid duplication, studies with similar sample sizes will not be used more than once, and in the case of similar studies, studies with the largest sample size will be considered. Moreover, all studies conducted in regions other than sub-Saharan Africa will be excluded from this study.

## Search strategies for identification of relevant studies

The primary search strategies for electronic databases will include PubMed/MEDLINE, Scopus, EMBASE, PsycINFO, and the Web of Science for studies reporting primary data on the prevalence of perinatal suicidal behavior in East Africa. A combination of the medical subject headings, Boolean operators, and keywords will be used as the search terms. The search term will include MESH terms and free text variations determined for each database (S1 Appendix).

The other search strategy will be done with Google Scholar, using a combination of the medical subject headings and keywords as the search terms. Moreover, Google Scholar will also be searched for relevant grey literature, and reference lists of the included studies will be manually searched for further relevant literature. The search term will include perinatal suicidal behavior, perinatal suicidal ideation, suicidal attempt, suicidal gesture, suicidal pregnant women, antenatal, postnatal, horn of Africa, sub-Saharan Africa, sub-Saharan African countries, etc.

## Screening procedure

Two independent evaluators (JE and AM) will each conduct a preliminary screening process. To validate that the studies fit the inclusion requirements, authors will first examine the titles and abstracts before writing the contents. If there is a debate over whether to include an article, a third reviewer will make the decision. Mendeley software will be employed to organize references for this selection and omit duplicate articles. In order to prevent influence on the decision-making process, the screening will be conducted without any intervention or communication amongst reviewers.

## Data extraction

Two independent reviewers (AM and DN) will extract the data from eligible studies. The extracted data will be prepared in Microsoft Excel spreadsheets. The identified studies will be exported into the reference citation manager software to remove the duplicates. The two reviewers (AD and WG) will independently review the titles and abstracts of the studies. The studies will be classified into included, excluded, and undecided categories. The two authors will again independently assess the full texts of the included and undecided categories of studies against the eligibility criteria to include them in the final group of studies. The studies will be judged based on the eligibility criteria. Justification for the excluded studies will be described. In case of any disagreement raised among reviewers, it will be solved through communication, discussion, and inviting the third reviewer (KJ). The other four authors will abstract data systematically using the data extraction form. The data extraction form will contain the type of study, study subject characteristics, outcomes of interest, contextual factors, and other reported important factors and findings on the measures of association. The reviewers will contact the authors of the article and request details through email in the case of missing data or an incomplete report.

## Data synthesis

The extracted data from the eligible studies will be imported into STATA version 15 for analysis. Pertinent data extracted from all the eligible studies will be organized in the form of tables. A flow chart will be provided to show the methodological procedures of the study. Tables will be used to illustrate the parameters and quality rating of the included studies. Forest plots will be used to display compiled estimations. A meta-analysis of the prevalence of suicidal behavior will be conducted using a random-effects model, which will generate a pooled prevalence with respective 95% CIs. The pooled prevalence of suicidal behavior will be estimated by categorizing the population into different groups. For example, age, location or country, antepartum or postpartum period, year of publication, tools used, types of study, methodological quality, etc.). Heterogeneity will be checked by using the $I^2$ test (Cochran Q test) at a p-value of 0.05. Heterogeneity will be assumed if the $I^2$ value is greater than 50%. Publication bias will be checked using a funnel plot, Beg's rank, and Eggers linear statistical tests [50]. A separate subgroup analysis by age, setting, pregnancy status, year of study, quality of study, and sample size will be carried out based on the data extracted. A sensitivity test will be carried out. The result of the review will be reported according to the PRISMA guidelines for reporting.

## Confidence in cumulative evidence

The quality of the evidence on the outcomes of interest and its strength of recommendation will be described using the GRADE (Grading of Recommendations Assessment, Development, and Evaluation) approach system. Accordingly, the findings will be classified as high, moderate, low, or very low quality. For instance, the evidence based on RCTs will be classified as high-quality evidence, whereas the evidence based on observational studies will be considered low-quality evidence [51].

## Discussion

Suicidal behavior assessment in the context of the perinatal era has to be considered a priority in public health, and there is no question that any mental health policies cannot be effectively implemented without sufficient information about the actual situation. There is a dearth of data regarding suicide behavior in the perinatal period in the Sub-Saharan region that can provide an overall assessment of the situation in the region over time. As far as the authors are aware, there has been no comprehensive review published to date depicting the scope of suicidal behavior and its determinants among perinatal women in sub-Saharan African countries. In order to better understand the pooled prevalence of suicide behavior and its determinants, both published and unpublished studies will be included in the investigation. Hence, we will use a systematic review to estimate the prevalence of suicidal behavior in the antenatal and postpartum periods in sub-Saharan Africa. Furthermore, to enrich our estimation, we also intended to assess the prevalence of suicidal behavior among women in both periods (antenatal and postpartum) by contrasting them with those who did not have any other specific conditions. In light of this, it will be essential to gather and synthesize empirical data for this study on suicidal behavior during the perinatal period. Doing so will assist in improving the evidence for planning various types of interventions in the specific context of suicidal behavior during the perinatal period. The findings of this review will be presented at academic conferences, meetings, and panel discussions. Moreover, it will be published in a journal with a strong international reputation.

Concerning the study's limitations, including studies published only in English and excluding articles published in other languages, there may be bias in the study's outcome.

## Conclusion

Systematically reviewing the literature on perinatal suicide will be expected to generate substantial evidence for future intervention, giving a way to combat the hidden face of maternal morbidity and mortality in the region. It will also help identify determinants affecting perinatal suicidal behavior when formulating policies and guidelines by informing different stakeholders involved in decision-making.

## Supporting information

**S1 Checklist. PRISMA-P (Preferred Reporting Items for Systematic review and Meta-Analysis Protocols) 2015 checklist: Recommended items to address in a systematic review and meta-analysis protocol\*.**
(DOC)

**S1 Appendix. Literature search strategy.**
(DOCX)

## Author Contributions

**Conceptualization:** Wubishet Gezimu.

**Data curation:** Mohammedamin Hajure, Jemal Ebrahim, Aman Mamo.

**Formal analysis:** Gebiso Roba, Mustefa Mohammedhussein.

**Investigation:** Mohammedamin Hajure.

**Methodology:** Wubishet Gezimu, Desalegn Nigatu, Aman Dule, Kiyar Jemal.

**Project administration:** Mohammedamin Hajure.

**Software:** Gebiso Roba.

**Supervision:** Mustefa Mohammedhussein.

**Writing – original draft:** Mohammedamin Hajure.

**Writing – review & editing:** Mohammedamin Hajure, Wubishet Gezimu, Desalegn Nigatu, Aman Dule, Kiyar Jemal.

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
