## [Editor Report · Decision Letter 0]

20 Dec 2022

PONE-D-22-22410Perinatal suicidal behavior in Sub-Saharan Africa: A study protocol for a systematic review with meta-analysisPLOS ONE

Dear Dr. Hajure,

Thank you for submitting your manuscript to PLOS ONE. After careful consideration, we feel that it has merit but does not fully meet PLOS ONE’s publication criteria as it currently stands. Therefore, we invite you to submit a revised version of the manuscript that addresses the points raised during the review process.

We look forward to receiving your revised manuscript.

Kind regards,

María Eugenia Gómez López, Ph.D.

Academic Editor

PLOS ONE

Journal Requirements:

https://journals.plos.org/plosone/s/fileid=ba62/PLOSOne_formatting_sample_title_authors_affiliations.pdf.

2. PLOS ONE does not copy edit accepted manuscripts (https://journals.plos.org/plosone/s/criteria-for-publication#loc-5). To that effect, please ensure that your submission is free of typos and grammatical errors.

4. We note that this manuscript is a systematic review or meta-analysis; our author guidelines therefore require that you use PRISMA guidance to help improve reporting quality of this type of study. Please upload copies of the completed PRISMA checklist as Supporting Information with a file name “PRISMA checklist”.

Additional Editor Comments (if provided):

PONE-D22-22410 is a very interesting work, the topic is relevant not only for the African reasearch, but also for the perinatal mental health research in the world. However, the authors must consider the following. After a preliminary review of the submission, following the publication criteria of PLOS ONE, based on the AMSTAR and PRISMA checklists systematic reviews, it is my assessment:

According to AMSTAR guidelines, the following items are missing from the article:

1. Research question(s)

2. Objective (s) of the systematic review

3. Inclusion criteria must include: population, intervention, and comparator group (PICO)

4. Explanation of the selection of the study designs (If the authors will include RCTs, NRSI or both).

Based on PRISMA CHECKLIST (S1 Table), the following items are missing from the article:

1. Administrative information: identification and registration (must be on page 3 instead of 4).

2. Introduction: Objectives are missing and also the explicit statement of the question(s) the review will address according to PICO.

3. Methods: in the eligibility criteria are missing the study characteristics: PICO, study design, time frame and report characteristics. And also are missing the items such as: Outcomes and priorization, Meta-bias, and Confidence in cumulative evidence.

These resuIts of the preliminary review have led me to believe it is necessary to have the author work on the missing items mentioned above to complete their protocol before any reviewers can be invited to provide their assessment.

Best regards,

María Eugenia Gómez López, PhD

Academic Editor
---

## [Author Response · Author response to Decision Letter 0]

5 Jan 2023

Manuscript title: Perinatal suicidal behavior in Sub-Saharan Africa: A study protocol for a systematic review with meta-analysis

Manuscript ID No: PONE-D22-22410

Dear esteemed Editor, We are so grateful for your quick and positive response to our article. We appreciate your considerate and intellectual comments and suggestions. We addressed each of the comments and suggestions point-by-point as mentioned hereunder. 

In my opinion, PONE-D22-22410 is a very interesting work, the topic is relevant not only for the African research, but also for the perinatal mental health research in the world. However, the authors must consider the following. After a preliminary review of the submission, following the publication criteria of PLOS ONE, based on the AMSTAR and PRISMA checklists systematic reviews, it is my assessment:

According to AMSTAR guidelines, the following items are missing from the article:

1. Research question(s)

Authors’ response: Dear editor, thank you so much for your intellectual suggestions. Accordingly we included the research question in the revised version of the manuscript.

2. Objective (s) of the systematic review

Authors’ response: Thanks in advance. We have included the objective of the study in the last section of the introduction of the revised manuscript.

3. Inclusion criteria must include: population, intervention, and comparator group (PICO)

Authors’ response: Dear Editor! Thanks a lot for your insightful suggestion. We included the PICO search guideline under the eligibility criteria. Thanks once again. 

4. Explanation of the selection of the study designs (If the authors will include RCTs, NRSI or both). 

Authors’ response: Dear editor, we detailed mentioned the types studies to be included (in terms of study design) in the eligibility criteria. In the review we will consider all quantitative observational studies (cross-sectional, case-control, cohort, or longitudinal studies, and survey findings) and randomized control trial (RCT) studies will be included.

Based on PRISMA CHECKLIST (S1 Table), the following items are missing from the article:

1. Administrative information: identification and registration (must be on page 3 instead of 4).

 Authors’ response: Thank you so much for the recommendation. We have edited accordingly.

2. Introduction: Objectives are missing and also the explicit statement of the question(s) the review will address according to PICO.

Authors’ response: Thank you for your constructive comments. We have incorporated the comments in the main document. 

3. Methods: in the eligibility criteria are missing the study characteristics: PICO, study design, time frame and report characteristics. And also are missing the items such as: Outcomes and priorization, Meta-bias, and Confidence in cumulative evidence.

Authors’ response: Dear editor, thank you for your constructive comments. We have incorporated all the comments in the main document.

Journal Requirements:

Authors’ response: We ensured that our manuscript meets PLOS ONE's style requirements.

2. PLOS ONE does not copy edit accepted manuscripts (https://journals.plos.org/plosone/s/criteria-for-publication#loc-5). To that effect, please ensure that your submission is free of typos and grammatical errors.

 Authors’ response: We have checked and then corrected all typos and grammatical errors

Authors’ response: Thanks. Since our article is a study protocol, we stated the Data Availability as “The identified research data will be made publicly available when the study is completed and published”, as per the PLOS ONE’s Data Availability statement for protocol. We updated the Data Availability in the cover letter.

4. We note that this manuscript is a systematic review or meta-analysis; our author guidelines therefore require that you use PRISMA guidance to help improve reporting quality of this type of study. Please upload copies of the completed PRISMA checklist as Supporting Information with a file name “PRISMA checklist”.

Authors’ response: Thanks a lot for your suggestion. We have uploaded the PRISMA checklist.

---

## [Decision Letter · Decision Letter 1]

3 Apr 2023

PONE-D-22-22410R1Perinatal suicidal behavior in Sub-Saharan Africa: A study protocol for a systematic review with meta-analysisPLOS ONE

Dear Dr. Hajure,

Thank you for submitting your manuscript to PLOS ONE. After careful consideration, we feel that it has merit but does not fully meet PLOS ONE’s publication criteria as it currently stands. Therefore, we invite you to submit a revised version of the manuscript that addresses the points raised during the review process.

Reviewer 1

1) Does suicidal behavior include suicidal ideation? right now this point is unclear, although the authors included suicidal ideation in the search strategy and defined suicidal ideation, it is not covered in the definition of suicidal behaviors defined in Definition of concepts.

2) For eligibility criteria, why exclude studies published before the year 2000?

3) The description of the search strategy is somewhat hard to follow. Can the authors list the main search strategy with the key terms and then report the variations used for each key term? or if the authors have other better ways to present the search strategy it will be acceptable as long as easy to follow.

Reviewer 2

**Abstract:**

In the context of maternal morbidity, the ICD-10 classifies suicide as a direct cause of death, not an indirect cause of death as stated here. Please review same.

"Perinatal mental illnesses" are mental illnesses that exist during the perinatal period so this statement is misleading as it conveys a sense that they are separate, when they are not. Please revise for clarity.

The perinatal period alone does not increase susceptibility to suicidal behaviour. Again, this is misleading and should be reviewed.

For the abstract, there is too much detail provided in the methods section and a synopsis only should be provided.

**Introduction:**

Again perinatal mental illnesses are mental illnesses that exist during the perinatal period so this statement is misleading as it conveys a sense that they are separate, when they are not. Please revise for clarity.

Suicide is a symptom and not a condition. Please revise for clarity.

"Suicidal behaviour WAS thought to be a leading..." if this is past-tense, what is currently thought?

Again, the perinatal period alone does not increase susceptibility to suicidal behaviour. This is nuanced and should be reviewed.

Again, in the context of maternal morbidity, the ICD-10 classifies suicide as a direct cause of death, not an indirect cause of death as stated here. Please review same.

"During the perinatal period women have..." - this is not reflective of all countries and is too nuanced to be this clear-cut. There is evidence that despite women having regular contact with HCP's, women's mental health needs are not addressed and therefore, this needs to be reviewed to reflect this.

**Specific objectives:**

Please define what is meant by 'level'

**Specific questions:**

Please define what is meant by 'determines it'

**Methods:**

Please provide rationale for choosing 21 year timeframe.

Psych Info - please amend to read PsycINFO

Please consider language in search terms "committed" is no longer an accepted term in the context of suicide.

**Screening procedure:**

Please consider an alternative phrase to "get rid of" (omit, remove etc.)

**Discussion**

It is accepted little evidence exists in terms of sub-Saharan Africa, however, several reviews have been conducted on the prevalence rates and psychosocial risk factors relating to perinatal suicide/suicidal ideation or suicidal behaviours - please provide commentary on the state of the evidence currently and what gaps this review aims to address.

**Limitations**

Please ensure all language is future-tense, such as English language WILL be included as opposed to was included. Also I would query the line "in most studies, the death of the mother..." this reads as though the study has been completed so it is important not to confuse the protocol with the findings of the review.

**General:**

There are issues with the reference list - please ensure to address these.

We look forward to receiving your revised manuscript.

Kind regards,

María Eugenia Gómez López, Ph.D.

Academic Editor

PLOS ONE

Journal Requirements:

Reviewers' comments:

Reviewer's Responses to Questions

**Comments to the Author**

1. Does the manuscript provide a valid rationale for the proposed study, with clearly identified and justified research questions?

Reviewer #1: Yes

Reviewer #2: Yes

2. Is the protocol technically sound and planned in a manner that will lead to a meaningful outcome and allow testing the stated hypotheses?

Reviewer #1: Yes

Reviewer #2: Yes

3. Is the methodology feasible and described in sufficient detail to allow the work to be replicable?

Reviewer #1: Yes

Reviewer #2: No

4. Have the authors described where all data underlying the findings will be made available when the study is complete?

Reviewer #1: Yes

Reviewer #2: Yes

5. Is the manuscript presented in an intelligible fashion and written in standard English?

Reviewer #1: Yes

Reviewer #2: Yes

6. Review Comments to the Author

You may also provide optional suggestions and comments to authors that they might find helpful in planning their study.

Reviewer #1: This is a study protocol for a systematic review with meta-analysis on perinatal suicidal behavior in Sub-Saharan Africa. Perinatal psychiatric illness in particular suicidal behavior is a critical condition and has substantial influences on the development of the fetus and child. A systematic and meta analytic approach will help synthesize the evidence of previous studies and provide important insights that can guide future investigations. In general, I find the manuscript well written in an intelligible fashion and the authors have adequately addressed the editor and reviewers' concerns. I have three additional comments for the authors to address, which may help further enhance the manuscript.

1), does suicidal behavior include suicidal ideation? right now this point is unclear, although the authors included suicidal ideation in the search strategy and defined suicidal ideation, it is not covered in the definition of suicidal behaviors defined in Definition of concepts.

2), for eligibility criteria, why exclude studies published before the year 2000?

3), the description of the search strategy is somewhat hard to follow. Can the authors list the main search strategy with the key terms and then report the variations used for each key term? or if the authors have other better ways to present the search strategy it will be acceptable as long as easy to follow.

Reviewer #2: Authors,

Thank you for compiling this review protocol for a very worthy subject matter. My comments are intended to enhance and improve the paper.

Q3 above - As you have chosen to undertake a search of grey literature and unpublished studies, it would be difficult to replicate this. Therefore, I have answered 'no' to this question.

Abstract:

In the context of maternal morbidity, the ICD-10 classifies suicide as a direct cause of death, not an indirect cause of death as stated here. Please review same.

"perinatal mental illnesses" are mental illnesses that exist during the perinatal period so this statement is misleading as it conveys a sense that they are separate, when they are not. Please revise for clarity.

The perinatal period alone does not increase susceptibility to suicidal behaviour. Again, this is misleading and should be reviewed.

For the abstract, there is too much detail provided in the methods section and a synopsis only should be provided.

Introduction:

Again perinatal mental illnesses are mental illnesses that exist during the perinatal period so this statement is misleading as it conveys a sense that they are separate, when they are not. Please revise for clarity.

Suicide is a symptom and not a condition. Please revise for clarity.

"Suicidal behaviour WAS thought to be a leading..." if this is past-tense, what is currently thought?

Again, the perinatal period alone does not increase susceptibility to suicidal behaviour. This is nuanced and should be reviewed.

Again, in the context of maternal morbidity, the ICD-10 classifies suicide as a direct cause of death, not an indirect cause of death as stated here. Please review same.

"During the perinatal period women have..." - this is not reflective of all countries and is too nuanced to be this clear-cut. There is evidence that despite women having regular contact with HCP's, women's mental health needs are not addressed and therefore, this needs to be reviewed to reflect this.

Specific objectives:

Please define what is meant by 'level'

Specific questions:

Please define what is meant by 'determines it'

Methods:

Please provide rationale for choosing 21 year timeframe.

Psych Info - please amend to read PsycINFO

Please consider language in search terms "committed" is no longer an accepted term in the context of suicide.

Screening procedure:

Please consider an alternative phrase to "get rid of" (omit, remove etc.)

Discussion

It is accepted little evidence exists in terms of sub-Saharan Africa, however, several reviews have been conducted on the prevalence rates and psychosocial risk factors relating to perinatal suicide/suicidal ideation or suicidal behaviours - please provide commentary on the state of the evidence currently and what gaps this review aims to address.

Limitations

Please ensure all language is future-tense, such as English language WILL be included as opposed to was included. Also I would query the line "in most studies, the death of the mother...." this reads as though the study has been completed so it is important not to confuse the protocol with the findings of the review.

General:

There are issues with the reference list - please ensure to address these.

7. PLOS authors have the option to publish the peer review history of their article (what does this mean?). If published, this will include your full peer review and any attached files.

Reviewer #1: No

Reviewer #2: No

---

## [Author Response · Author response to Decision Letter 1]

17 Apr 2023

Response to Reviewers Comments

Manuscript ID: PONE-D-22-22410R1

Title: Perinatal suicidal behavior in Sub-Saharan Africa: A study protocol for a systematic review with meta-analysis

Dear Editor and Reviewers: Thank you so much for your time with our manuscript and your constructive comments and suggestions, which greatly helped in revising our protocol. We accepted all the comments and suggestions and amended the manuscript accordingly. The point-by-point responses to the reviewers’ comments are listed below.

Reviewer 1

1) Does suicidal behavior include suicidal ideation? right now this point is unclear, although the authors included suicidal ideation in the search strategy and defined suicidal ideation, it is not covered in the definition of suicidal behaviors defined in Definition of concepts.

Authors’ response: Dear Editor, thank you so much for your intellectual suggestions. Frankly speaking, suicidal behavior does not include suicidal ideation. We thought this could be explained by some screening instruments used only for research purposes. However, for the current review, we planned to state that it was suicidal behavior. We will accept reviewer comments to modify and include suicidal ideation in the suicidal behavior as follows: ‘‘Suicidal behavior- includes suicidal ideation, intent, thought attempt, and completion." So we have presented a reference to an earlier review article as evidence. (Kessler and Ustun, 2004).

2) For eligibility criteria, why exclude studies published before the year 2000?

Authors’ response: Thank you so much for your comments. Actually, we don’t have a scientific basis for this issue. Authors have been interested in observing the trend over the past 20 years, with the rough assumption that most of the studies conducted on suicidal behavior are at least 20 years old. However, we have accepted and excluded time limitations. Thanks again.

3) The description of the search strategy is somewhat hard to follow. Can the authors list the main search strategy with the key terms and then report the variations used for each key term? or if the authors have other better ways to present the search strategy it will be acceptable as long as easy to follow.

Authors’ response: Thank you so much for your comments. We have addressed the comment and uploaded the database search strategy as supplementary information. See supplementary information 1.

Reviewer 2

Abstract:

In the context of maternal morbidity, the ICD-10 classifies suicide as a direct cause of death, not an indirect cause of death as stated here. Please review same.

Authors’ response: Thank you so much for your intellectual suggestion. We have amended the mentioned section as per your information. 

"Perinatal mental illnesses" are mental illnesses that exist during the perinatal period so this statement is misleading as it conveys a sense that they are separate, when they are not. Please revise for clarity.

Authors’ response: Thanks for your suggestion. We have revised the mentioned section. 

The perinatal period alone does not increase susceptibility to suicidal behaviour. Again, this is misleading and should be reviewed.

Authors’ response: Thanks a lot for your suggestion. We have revised the suggested sentence.

For the abstract, there is too much detail provided in the methods section and a synopsis only should be provided.

Authors’ response: Thanks. We have revised the mentioned section accordingly. 

Introduction:

Again perinatal mental illnesses are mental illnesses that exist during the perinatal period so this statement is misleading as it conveys a sense that they are separate, when they are not. Please revise for clarity.

Authors’ response: Thanks for your suggestion. We have revised the mentioned section. 

Suicide is a symptom and not a condition. Please revise for clarity.

Authors’ response: Thanks a lot. We have revised the mentioned word. 

"Suicidal behaviour WAS thought to be a leading..." if this is past-tense, what is currently thought?

Authors’ response: Thanks a lot. We have revised the mentioned tense.

Again, the perinatal period alone does not increase susceptibility to suicidal behaviour. This is nuanced and should be reviewed.

Authors’ response: Thanks a lot. We have revised the mentioned section. 

Again, in the context of maternal morbidity, the ICD-10 classifies suicide as a direct cause of death, not an indirect cause of death as stated here. Please review same.

Authors’ response: Thanks you so much. We have revised the mentioned information accordingly. 

"During the perinatal period women have..." - this is not reflective of all countries and is too nuanced to be this clear-cut. There is evidence that despite women having regular contact with HCP's, women's mental health needs are not addressed and therefore, this needs to be reviewed to reflect this.

Authors’ response: Thank you so much your intellectual suggestions. We have revised the mentioned paragraph. 

Specific objectives

Please define what is meant by 'level'

Authors’ response: Thanks a lot. It was to mean “prevalence”. We have replaced it with “prevalence” in the revised version.

Specific questions:

Please define what is meant by 'determines it'

Authors’ response: Thanks for your suggestion. It was to mean “what determines that prevalence”

Methods:

Please provide rationale for choosing 21 year timeframe.

Authors’ response: Thanks for your intellectual suggestion. Actually, we have no any solid rationale for choosing 21 year timeframe for our review rather the researchers (our) interest. We have unlimited the publication year depending on your suggestion. Thanks a lot.

Psych Info - please amend to read PsycINFO

Authors’ response: Thanks a lot. We have amended the mentioned section accordingly.

Please consider language in search terms; "committed" is no longer an accepted term in the context of suicide.

Authors’ response: Thank you so much for your intelligent suggestion. We will not consider "committed" in the search terms. 

Screening procedure:

Please consider an alternative phrase to "get rid of" (omit, remove etc.)

Authors’ response: Thank you so much. We replaced the phrase "get rid of" with “omit” as per your comment. Thanks

Discussion

It is accepted little evidence exists in terms of sub-Saharan Africa, however, several reviews have been conducted on the prevalence rates and psychosocial risk factors relating to perinatal suicide/suicidal ideation or suicidal behaviours - please provide commentary on the state of the evidence currently and what gaps this review aims to address.

Authors’ response: Dear Editor! Thanks a lot for your constructive suggestions.

Suicide remains the leading cause of direct maternal death in the first postnatal year (Saving Lives, Improving Mothers’ Care 2021 report), and it is among the top three causes of pregnancy-associated deaths, as defined by the CDC (Lindahl, V., Pearson, J. L., & Colpe, L. (2005). Some studies have also shown black women are twice as likely as white women to report having suicidal thoughts (suicidal ideation) in the immediate postpartum period (doi: 10.1089/jwh.2015.5346). In terms of causes of adolescent deaths, suicide ranked third globally and first in Africa (11.2 per 100 000 people) (WHO, 2021).

Considering an earlier review on suicidal ideation and behavior in low and middle-income countries, two (2) studies have been done so far, and the differential explanation is presented below. A systematic review conducted by Daniela C. Fuhr et al. (2014) focused on the identification of the studies reporting the proportion of pregnancy-related deaths from injuries and suicide in low- and middle-income countries. So this study focused only on completed types of suicide and injuries from related psychosocial risk factors, in contrast to the current study, which focuses on all types of suicidal forms, which include suicidal ideation, intent, gesture, and attempt, as well as suicide deaths. Secondly, the a review by Lindahl V, Pearson JL, and Colpe L. (2005) was done worldwide, but only single studies in South Africa were included from Africa. As stated earlier, it is quite different compared to the current one in terms of study setting (continent: worldwide vs. sub-Saharan), the timing of suicide (pregnancy and postpartum vs. antepartum, pregnancy, and postpartum), and timing (included study before 2005 E.C.). Furthermore, the Joan M et al. (2022) study excluded suicide and focused on the perinatal mental health of pregnant teenagers in sub-Saharan Africa. The results of this study (Joan M et al., 2012) and the other reviews discussed above may have shed some light on the concept. However, it is improbable that a clear correlation between suicidal ideation and behavior and a significant psychosocial factor affecting overall health has been identified in the region.

So the assessment of suicide in the context of maternal health, in particular during the perinatal period, in sub-Saharan African countries needs to be considered a public health priority, and there is no doubt that any mental health policies could not be efficiently implemented without adequate evidence of the situation on the ground. Therefore, this study will be imperative to collect and combine empirical data on suicide among mothers in the perinatal period, and doing so will help to provide better evidence to plan psycho-social interventions in the particular context of maternal mental health.

Limitations

Please ensure all language is future-tense, such as English language WILL be included as opposed to was included. Also I would query the line "in most studies, the death of the mother..." this reads as though the study has been completed so it is important not to confuse the protocol with the findings of the review.

Authors’ response: Thank you so much for your suggestion. We have corrected all the tense errors the mentioned section accordingly. 

General:

There are issues with the reference list - please ensure to address these.

Authors’ response: Thanks a lot. We have checked and revised all the errors in the references list.

Journal Requirements:

Author response: Thank you so much for your important suggestion. We have checked the reference list for its completeness and correctness. We have not cited any retracted articles. Moreover, we updated reference lists 8 and 22. Moreover, we have checked and corrected all errors in the references list.

---

## [Editor Report · Decision Letter 2]

24 Apr 2023

Perinatal suicidal behavior in Sub-Saharan Africa: A study protocol for a systematic review with meta-analysis

PONE-D-22-22410R2

Dear Dr. Hajure:

We’re pleased to inform you that your manuscript has been judged scientifically suitable for publication and will be formally accepted for publication once it meets all outstanding technical requirements.

Kind regards,

María Eugenia Gómez López, Ph.D.

Academic Editor

PLOS ONE
---

## [Editor Report · Acceptance letter]

27 Apr 2023

PONE-D-22-22410R2 

Perinatal suicidal behavior in Sub-Saharan Africa: A study protocol for a systematic review with meta-analysis 

Dear Dr. Hajure:

I'm pleased to inform you that your manuscript has been deemed suitable for publication in PLOS ONE. Congratulations! Your manuscript is now with our production department. 

Kind regards, 

on behalf of

Dr. María Eugenia Gómez López 

Academic Editor

PLOS ONE